# Comparative Effects of Abdominal Bracing and Valsalva Maneuver on Cerebral and Peripheral Hemodynamics in Healthy Adults: A Randomized Crossover Study

**DOI:** 10.3390/medicina61112031

**Published:** 2025-11-14

**Authors:** Ji-Hyeon Yu, Ju-Yeon Jung, Yeong-Bae Lee, Jeong-Min Shim, Young-Don Son, Jiwon Yang, Chang-Ki Kang

**Affiliations:** 1Department of Neuroscience, Gachon Advanced Institute for Health Sciences & Technology (GAIHST), Gachon University, Incheon 21999, Republic of Korea; wlgus21@gachon.ac.kr (J.-H.Y.);; 2Institute of Human Convergence Health Science, Gachon University, Incheon 21936, Republic of Korea; 9955me@gachon.ac.kr; 3Department of Neurology, Gil Medical Center, College of Medicine, Gachon University, Incheon 21565, Republic of Korea; yeongbaelee@gachon.ac.kr; 4Department of Biomedical Engineering, College of IT Convergence, Gachon University, Seongnam 13120, Republic of Korea; ydson@gachon.ac.kr; 5Department of Radiological Science, College of Medical Science, Gachon University, Incheon 21936, Republic of Korea

**Keywords:** isometric abdominal activation, forced exhalation, intra-abdominal pressure, intra-thoracic pressure, cerebrovascular circulation, peripheral circulation

## Abstract

*Background and Objectives*: Blood flow is critical for tissue oxygenation, and alterations in cerebrovascular and peripheral circulation have important health implications. This study aimed to examine the impact of distinct mechanisms for increasing intra-cavity pressure through the abdominal bracing (AB) and Valsalva maneuver (VM) on central and peripheral hemodynamics. *Materials and Methods*: A randomized crossover design was used, and thirty healthy young adults (age 21.9 ± 1.5 years; BMI 20.9 ± 1.8 kg/m^2^) performed AB and VM in a randomized order. All participants provided written informed consent, and the study protocol was approved by the Clinical Research Information Service (KCT0009742; registered on 30 August 2024). Hemodynamic responses were measured before and after each intervention, including heart rate, blood pressure, pulse wave velocity, carotid artery diameter, pulsatility index, resistive index, peripheral oxygen saturation, and cerebral oxygenation. Repeated-measures analysis of variance and paired *t*-tests were conducted on the datasets. *Results*: Both the VM and AB significantly increased heart rate (*p* < 0.001) and systolic blood pressure (VM: *p* = 0.015; AB: *p* < 0.001). Cerebral oxygen saturation decreased significantly (VM: *p* < 0.05; AB: *p* < 0.05), whereas oxyhemoglobin increased during both interventions, suggesting higher cerebral oxygen demand. The VM specifically increased the carotid pulsatility index (pre = 1.76 ± 0.28; post2 = 1.87 ± 0.33; *p* = 0.008), reflecting elevated central vascular resistance. In contrast, AB decreased peripheral oxygen saturation (pre = 98.43 ± 0.71; post1 = 97.49 ± 1.76; *p* < 0.001) and increased peripheral (heart–finger) pulse wave velocity (Lt: *p* = 0.026; Rt: *p* = 0.010), indicating greater stimulation of peripheral circulation. *Conclusions*: Distinct mechanisms that elevate intra-cavity pressure differentially influence central and peripheral hemodynamics. These findings suggest that intra-cavity pressure can selectively modulate hemodynamic responses, with potential applications in both clinical and exercise settings.

## 1. Introduction

Systemic circulation begins in the aorta and branches into large arteries that narrow into smaller vessels [1]. Capillaries are the terminal vessels of the arterial system, delivering oxygen and nutrients to organs, tissues, and cells. Vessels branching from the aorta are broadly divided into peripheral and cerebral circulation. Peripheral circulation is crucial for maintaining tissue function and muscle activity. Reduced peripheral blood flow can cause muscle pain, numbness, and cramps [2,3]. The incidence of peripheral arterial disease rises sharply with age and is linked to diabetes, hyperlipidemia, smoking, and other risk factors. Peripheral arterial disease (PAD), affecting more than 200 million individuals globally, shows a substantial association with hypertension, with approximately 17.0% of cases attributable to elevated blood pressure in a large epidemiological study [2,4]. It can progress to long-term complications such as ischemia and sepsis [2,5,6].

Cerebral blood flow (CBF) represents approximately 15% of cardiac output and sustains neural activity by supplying oxygen and nutrients continuously [7]. Reduced CBF can lead to neurological decline and irreversible brain damage [8]. Early evaluation and intervention are therefore essential for disease prevention and health promotion, as reduced blood flow is not merely transient but can cause neurological injury and life-threatening systemic complications. Blood circulation is regulated by arterial elasticity and blood pressure, which are essential for maintaining adequate flow. CBF increases in proportion to cerebral perfusion pressure, itself modulated by changes in mean arterial pressure. Peripheral blood flow is also influenced by blood pressure through vessel diameter changes mediated by sympathetic activation. Thus, blood pressure is a key determinant of both CBF and peripheral circulation, and appropriate modulation of blood pressure can stimulate increased flow to the brain and peripheral tissues.

A common way to alter blood pressure is by increasing intrathoracic or abdominal pressure. Abdominal bracing (AB) and the Valsalva maneuver (VM) are representative methods for controlling intra-cavity pressure. The VM, in particular, maintains constant pressure during breath-holding and produces immediate physiological effects such as elevated blood pressure and reduced venous return. Similar respiratory patterns occur in daily activities like heavy lifting or coughing [9]. This approach rapidly increases blood pressure and alters cardiovascular and autonomic nervous system function through elevated intra-abdominal and intrathoracic pressure.

AB, similar to the VM, increases intra-abdominal and intrathoracic pressure through simultaneous contraction of abdominal and lumbar muscles. Unlike the VM, however, it is performed while maintaining respiration [10,11,12]. This maneuver elevates intra-cavity pressure and may elicit hemodynamic responses similar to the VM by raising blood pressure [13]. During the isometric contraction induced by AB, muscle contraction compresses blood vessels, altering blood flow and oxygen delivery. When the contraction is released, blood flow through previously compressed vessels increases rapidly, promoting vasodilation within skeletal muscle. Moreover, muscle activation and the resulting rise in oxygen consumption are supported by an enhanced blood supply [14].

On the other hand, there remains a lack of research addressing hemodynamic changes induced by intra-cavity pressure alterations such as those produced during the VM and AB. Prior studies have also reported VM-induced cerebral hemodynamic alternations mediated by blood pressure and physiological changes [9,15,16]; however, these investigations have predominantly focused on the carotid artery, leaving peripheral vascular responses insufficiently explored. Similarly, limited evidence exists regarding the hemodynamic effects of AB. Although AB shares a mechanistic similarity with VM, previous studies have largely centered on muscle activation and core stability, with hemodynamic influences remaining under investigated.

Therefore, the present study aimed to examine the impact of distinct mechanisms for increasing intra-cavity pressure through the VM and AB on central and peripheral hemodynamics to evaluate their physiological significance and potential clinical applicability. Notably, this study represents the first attempt to compare VM and AB from a hemodynamic perspective, highlighting its potential academic relevance. Based on this rationale, the study hypotheses were as follows. (1) VM and AB would affect the structure and function of either central or peripheral vessels. (2) Differences in pressure induction between VM and AB would elicit distinct effects on the structure and function of central and peripheral vessels.

## 2. Materials and Methods

### 2.1. Study Design

This study was conducted as a randomized crossover experimental study with reference to the Transparent Reporting of Evaluations with Nonrandomized Designs (TREND) checklist. The study was conducted in accordance with the Declaration of Helsinki, approved by the Institutional Review Board (approval date: 9 July 2024; approval number: 1044396-202405-HR-088-01), and registered in the Clinical Research Information System (date of registration: 30 August 2024; registration number: KCT0009742; patient enrollment and trial start date: 1 September 2024; trial end date: 31 December 2024). All participants provided written informed consent before beginning the study.

### 2.2. Participants

Thirty healthy adults in their twenties were included, meeting the inclusion criteria of having a normal body mass index (BMI) and blood pressure, and being able to generate expiratory pressures of 20–25 mmHg. The exclusion criteria included a history of diseases affecting CBF, abnormal neurological findings, disorders or medications related to cerebral dysfunction, or respiratory conditions. In addition, individuals with professional training or education related to exercise, or those engaged in high-intensity regular physical activity (such as strength training, aerobic exercise, or conditioning) were excluded. To minimize confounding, participants were instructed to avoid substances affecting the vascular or autonomic nervous systems (caffeine, alcohol, and smoking) for 24 h before testing and they took part in the experiments in a rest state. Participants with abnormal body mass index (BMI), blood pressure, or other indicators outside the normal range were also excluded to reduce cardiovascular confounding. BMI was calculated from baseline measurements of weight and height obtained using a calibrated scale (Atflee T3; Atflee Co., Ltd., Seoul, Republic of Korea) and a measuring tape, with height measured while participants stood barefoot, upright, and facing forward in the Frankfort plane [17]. The final sample consisted of 30 participants (mean age 21.9 ± 1.5 years; 15 females, 15 males; BMI 20.9 ± 1.8 kg/m^2^). To enhance participants’ adherence, small incentives were provided.

The required sample size was calculated using G*Power 3.1.9.7, based on partial η^2^ (≒0.07) and effect size (f = 0.28) for heart rate (HR) changes induced by the VM [18], derived from a pilot test conducted within this study. Repeated-measures analysis of variance with within-subject factors, power of 0.95, and significance level of 0.05 indicated a minimum of 26 participants. Accounting for a 10% dropout rate, 29 were required; ultimately, 30 were enrolled.

### 2.3. Data Acquisition

#### 2.3.1. Physiological Parameters Measurements

Physiological changes during the interventions were monitored with a patient monitor providing continuous real-time measurements of peripheral capillary oxygen saturation (SpO_2_) and HR. Systolic blood pressure (SBP) and diastolic blood pressure (DBP) were measured before and after the task using a smartwatch (SM-R850; Samsung Electronics Co., Ltd., Suwon, Republic of Korea). The smartwatch was worn on the left wrist and calibrated three times against an electronic sphygmomanometer (TMB-1112; Guangdong Transtek Medical Electronics Co., Ltd., Zhongshan, China) (Figure 1).

#### 2.3.2. Vascular Hemodynamic Parameters Measurements

To assess r espiration-related hemodynamic changes, the lumen diameter, pulsatility index (PI), and resistive index (RI) of the common carotid artery (CCA) were measured approximately 2 cm distal to the bifurcation of the right internal and external carotid arteries [19]. Measurements were obtained using a diagnostic ultrasound system (E-CUBE i7; Alpinion Medical Systems Co., Ltd., Seoul, Republic of Korea) with a 3–12 MHz linear transducer (L3-12T; Alpinion Medical Systems Co., Ltd., Seoul, Republic of Korea) (Figure 1). The preset of ultrasound measurements is as follows: a depth of 4.0 cm, dynamic range of 42 dB, gain of 36, wall filter of 2, and pulse repetition frequency (PRF) of 3.6 kHz. The CCA was chosen for its accessibility and established relevance in cerebral hemodynamic studies [20,21]. Furthermore, all ultrasound examinations were performed by an operator with more than four years of ultrasound research experience [15,22,23].

##### Carotid Pulse Wave Velocity (PWV) Measurements

Carotid PWV was used as a hemodynamic parameter to evaluate central arterial stiffness and elasticity. Vessel diameters for PWV calculation were obtained from brightness mode (B-mode) ultrasound images recorded over 5 s. The minimum (D_min_) and maximum (D_max_) lumen diameters were measured from these images using the intima-intima boundary tracking method in RadiAnt DICOM Viewer (version 2025.1; Medixant, 2025; https://www.radiantviewer.com (accessed on 12 September 2025); Figure 2A) [24]. PWV was then calculated with the Moens–Korteweg equation, assuming a blood density (ρ) of 1050 kg/m^3^ [25].

PWV was calculated as follows:$$PWV=\sqrt{\beta \times \frac{DBP\times 0.13333}{2\times \rho }}$$
where ρ = 1050 kg/m^3^.

The stiffness index *β* was calculated as follows:$$\beta (a.u.)=\frac{\ln SBP-\ln DBP}{(D_{max}-D_{min})/D_{min}}$$

##### Central Pulsed Doppler Measurements

To obtain additional hemodynamic data, pulsed Doppler mode was used to measure the PI and RI of the CCA in the longitudinal direction for 15 s before and 60 s after the respiratory intervention. To minimize error and improve accuracy, color Doppler was used to confirm the direction of blood flow, and the Doppler angle was fixed at 60° (Figure 2B,C). Additionally, a 1.0 mm sample volume was placed at the center of the long-axis carotid artery, and the angle correction cursor was aligned parallel to the vessel centerline. For consistency across sessions, the initial measurement site was marked to ensure an identical range of measurement.

##### Peripheral Pulse Wave Velocity Measurements

Peripheral arterial stiffness was indirectly assessed using heart–finger PWV, measured with an autonomic nervous system testing device (STD-1000; StraTek Co., Ltd., Seoul, Republic of Korea) (Figure 1). The method is based on pulse transit time and subject height (used to estimate arm length; height × 0.56). Four electrocardiogram electrodes were attached to both wrists and ankles, and photoplethysmography sensors were placed on the index fingers. Blood flow transit time was calculated from the delay between the electrocardiogram R-wave and photoplethysmography waveform, and PWV was then computed using the estimated heart–finger distance derived from participant height, providing an indirect measure of peripheral arterial stiffness.

#### 2.3.3. Cerebral Hemodynamic Parameters Measurements

To assess hemodynamic changes, near-infrared spectroscopy (NIRSIT ON; OBELAB Co., Inc., Seoul, Republic of Korea) was used to measure regional oxygen saturation (rSO_2_) and oxyhemoglobin (HbO) levels in the left and right prefrontal cortices (Figure 1). Near-infrared spectroscopy (NIRS) patches were placed above both eyebrows, and continuous real-time measurements were obtained for approximately 95 s, covering periods before and after the intervention.

### 2.4. Experimental Protocol

A prospective, controlled experimental study was carried out at the ultrasonography room, where all VM and AB interventions were performed in a controlled laboratory setting. Two interventions were carried out under the experimenter’s guidance, and participants were given instructions and practice for both the VM and AB. In this study, participants performed the VM with a mouthpiece held between the teeth, maintaining an expiratory pressure of 20–25 mmHg during a 15-s forced exhalation following a 5-s inspiration. Afterward, they returned to normal nasal breathing with the mouthpiece removed.

AB was performed in the same posture as the VM but without a mouthpiece. A pressure biofeedback device (Stabilizer; Chattanooga Group, Inc., Chattanooga, TN, USA) was placed at the lumbar curve between the first lumbar and second sacral vertebrae, allowing participants to generate pressure through core muscle activation. The device was connected to a pressure gauge, enabling real-time monitoring similar to the VM. During AB, participants contracted the abdominal muscles by firmly bracing the abdomen without moving the back or pelvis, while maintaining a steady breathing rhythm [26]. Intra-abdominal pressure was maintained at 20–25 mmHg above baseline for 15 s using the pressure biofeedback device [27].

Before the main experiment, participants rested in a hook-lying position with knees flexed at 90° for 10 min to stabilize physiological signals and allow baseline data collection (Figure 1). The two sessions of VM and AB were performed in random order (Figure 3). The order of assignment was randomized using the Excel random function, with 15 participants performing AB followed by VM, and 15 participants performing VM followed by AB. A washout period of 10 min was provided between the two interventions.

For each session, physiological parameters (SpO_2_, HR, blood pressure), hemodynamic parameters (vessel diameter, PI, RI, heart–finger PWV), and cerebral oxygenation parameters (rSO_2_, HbO) were measured before and after the intervention (Figure 1). The delivery of experimental instructions and the measurement of all variables were conducted in accordance with the single-blind procedure.

### 2.5. Data Analysis

For physiological, vascular, and cerebral hemodynamic parameters (including NIRS, PI, RI, SpO_2_, and HR), the 15 s before the intervention were defined as the pre-intervention period. The 60 s after the intervention were averaged in 15-s intervals and divided into four sections (post1–4) to analyze temporal changes. Repeated-measures analysis of variance with five within-subject factors was conducted. When the assumption of sphericity was violated, degrees of freedom were corrected using the ε-value: the Huynh–Feldt correction was applied when ε > 0.75, and the Greenhouse–Geisser correction when ε ≤ 0.75. The results were presented as the mean ± standard deviation (SD), F-values, and *p*-value for each variable, along with t-value, Tukey-adjusted *p*-value for each factor, and effect size.

To compare changes in PWV and stiffness index before and after the intervention, blood pressure and vessel diameter data were defined as pre and post, and analyzed with a paired *t*-test. The significance level was set at *p* < 0.05. Analyses were performed using Jamovi (version 2.3.28; https://www.jamovi.org, accessed on 2 October 2025). Results were presented as the mean ± standard deviation, mean difference ± standard error, t-value, *p*-value, and effect size. Statistics are reported to two decimal places, except mean differences and standard error differences, which are reported to three. Furthermore, *p*-values are reported as exact values (e.g., 0.05, 0.001) or as <0.001 where appropriate. Normality of all data was assessed using the Kolmogorov–Smirnov test, with *p* > 0.05 indicating a normal distribution.

## 3. Results

This study was conducted with thirty healthy participants. All participants were ethnically homogeneous and were university students. No participants withdrew from the experiment, and data from all thirty participants were included in the analysis.

### 3.1. Physiological Parameters

HR increased significantly from pre to post1 in both the VM (t = –3.73, *p* = 0.007) and AB (t = −5.08, *p* < 0.001), followed by a gradual decline toward baseline (VM: partial η^2^ = 0.24; AB: partial η^2^ = 0.41). Under VM, HR rose by 5.29 bpm, from 70.57 to 75.86 bpm, whereas under AB, it increased by 11.28 bpm, from 69.05 to 80.33 bpm (Appendix A). SpO_2_ showed a non-significant increasing trend during the VM (F = 0.02, *p* = 0.991, VM: partial η^2^ = 0.00). In contrast, AB produced a significant decrease (F = 5.56, *p* < 0.001, partial η^2^ = 0.16), with significance observed between pre and post1 (t = 3.17, *p* = 0.027), indicating reduced peripheral oxygen saturation during AB (Figure 4).

In the VM, SBP increased significantly by 1.33 mmHg, from 110.23 to 111.57 mmHg (t = −2.59, *p* = 0.015, Cohen’s *d* = −0.47), whereas DBP showed a non-significant upward trend of 0.77 mmHg, from 70.97 to 71.73 mmHg (t = −1.96, *p* = 0.060, Cohen’s *d* = −0.36; Appendix A). In contrast, during AB, both SBP and DBP increased significantly, by 2.03 mmHg and 1.63 mmHg, respectively (SBP: t = −4.25, *p* < 0.001, Cohen’s *d* = −0.78; DBP: t = −4.75, *p* < 0.001, Cohen’s *d* = −0.87).

### 3.2. Vascular Hemodynamic Parameters

Table 1 summarizes the statistical results for indices of vascular resistance. During the VM, the PI increased significantly from 1.76 at pre to 1.82 at post1 and 1.87 at post2 (F = 4.48, *p* = 0.008, partial η^2^ = 0.13). The RI also showed a significant change in repeated-measures analysis of variance (F = 2.87, *p* = 0.042, partial η^2^ = 0.09), although this was not confirmed in post hoc analysis (Appendix A). In contrast, AB produced no significant changes in either PI (F = 0.38, *p* = 0.763, partial η^2^ = 0.01) or RI (F = 0.32, *p* = 0.802, partial η^2^ = 0.01).

The hemodynamic parameters measured in the right CCA are presented in Appendix A. Both the VM and AB produced significant increases in vessel diameters (VM: D_min_, t = −2.33, *p* = 0.027, Cohen’s *d* = −0.42; D_max_, t = −3.68, *p* < 0.001, Cohen’s *d* = −0.67; AB: D_min_, t = −2.91, *p* = 0.007, Cohen’s *d* = −0.53; D_max_, t = −4.06, *p* < 0.001, Cohen’s *d* = −0.74). Carotid PWV decreased slightly but non-significantly, by 0.038 m/s in VM (from 4.61 to 4.53 m/s, Cohen’s *d* = 0.11) and by 0.039 m/s in AB (from 4.59 to 4.55 m/s, Cohen’s *d* = 0.05). Heart–finger PWV showed no significant change during the VM but increased significantly during AB (left: t = −2.35, *p* = 0.026, Cohen’s *d* = −0.43; right: t = −2.75, *p* = 0.010, Cohen’s *d* = −0.50). Specifically, AB increased heart–finger PWV by 2.856 cm/s on the left (from 424.96 to 427.31 cm/s) and by 1.797 cm/s on the right (from 426.62 to 428.42 cm/s).

### 3.3. Cerebral Hemodynamic Parameters

Cerebral hemodynamic signals measured with near-infrared spectroscopy are shown in Appendix A and Table 1. Notably, rSO_2_ decreased significantly in both the VM (left: F = 5.70, *p* = 0.005, partial η^2^ = 0.17; right: F = 10.99, *p* < 0.001, partial η^2^ = 0.29) and AB (left: F = 6.15, *p* = 0.001, partial η^2^ = 0.16; right: F = 5.95, *p* = 0.002, partial η^2^ = 0.18). Under VM, rSO_2_ in the left and right prefrontal cortices decreased by 4.25 and 5.44, respectively, immediately after the intervention (post1) compared with pre (left: t = 3.26, *p* = 0.023; right: t = 4.06, *p* = 0.003). Similarly, under AB, rSO_2_ in the left and right prefrontal cortices decreased by 3.49 and 3.74, respectively, at post1 (left: t = 3.47, *p* = 0.014; right: t = 3.36, *p* = 0.018). For HbO, the VM produced a significant increase in the left prefrontal cortex (F = 4.67, *p* = 0.012, partial η^2^ = 0.15) but not in the right (F = 1.99, *p* = 0.129, partial η^2^ = 0.07). Under AB, HbO increased significantly in both cortices (left: F = 6.63, *p* = 0.002, partial η^2^ = 0.20; right: F = 4.45, *p* = 0.014, partial η^2^ = 0.14), with a marked rise from pre to post1 in the left cortex (t = −2.95, *p* = 0.047).

## 4. Discussion

In this study, cerebral and peripheral hemodynamic changes during AB and the VM were analyzed. Both interventions produced common responses in HR, blood pressure, vessel diameters, and rSO_2_, although differences in underlying mechanisms and blood flow distribution were evident. The PI increased significantly only during the VM, whereas SpO_2_ and heart–finger PWV changed significantly only during AB (Figure 4).

The study found that the PI rose significantly after the VM. Because the PI reflects vascular resistance, lumen narrowing or increased blood flow velocity can elevate this parameter. During the VM, intrathoracic pressure rises and the sympathetic nervous system is activated, increasing vascular resistance. Upon release of intrathoracic pressure, venous return—previously impeded—is enhanced, further elevating the PI [28]. Similar findings have been reported, with vascular resistance rising immediately after the VM [15].

In contrast, AB did not significantly affect the PI. Unlike the VM, AB elevates intra-abdominal pressure while allowing respiration, limiting the degree of intrathoracic compression [28,29]. The VM, by comparison, combines increased intra-abdominal pressure with breath-holding, imposing greater mechanical stress on thoracic organs and directly affecting the heart and aorta. Because the PI in this study was measured in the CCA, which lies close to the aorta, it may have been particularly sensitive to the cardiovascular effects of the VM. Overall, both interventions elevated blood pressure, although PI changes in central arteries such as the CCA indicate that intrathoracic pressure from the VM had the stronger effect.

The effect on peripheral circulation was greater during AB. Significant changes in SpO_2_, measured at the index finger, were observed only in AB, with a marked decrease from pre to post1. This decline likely reflects increased oxygen consumption caused by whole-muscle contractions, including the abdominal muscles [30,31,32]. Unlike the VM, which engages abdominal muscles primarily through expiratory pressure regulation, AB recruits core muscles including the transversus abdominis, internal and external obliques, and rectus abdominis within a kinetic chain, acting as a corset that transmits force to the limbs. This trunk-stability exercise induces complex interactions among local, global, and load-transmitting muscles, thereby influencing limb muscle activity [33,34].

Consequently, adenosine triphosphate consumption rises, and oxidative phosphorylation is activated to resynthesize adenosine triphosphate, increasing oxygen demand and contributing to the observed reduction in SpO_2_. Simultaneously, the rise in intra-abdominal pressure and associated muscle contraction may compress vessels around the muscles, temporarily restricting regional blood flow and affecting oxygen delivery. In contrast, previous studies reported that SpO_2_ does not show marked fluctuations during the VM; instead, it gradually increases with intrathoracic pressure and returns slowly to baseline after completion. These findings support the interpretation that significant changes in peripheral oxygen saturation are unlikely during the VM [35,36].

Differences in peripheral hemodynamic responses between the two interventions were also evident in heart–finger PWV, with no significant changes detected during the VM, whereas AB produced a significant increase. The isometric contraction in AB likely induced systemic vasoconstriction, increasing vascular compression, partially restricting blood flow, and elevating blood pressure, which may have contributed to the transient rise in PWV immediately after the intervention [26,37]. Following release of vascular compression, vasodilation and enhanced blood flow could reduce peripheral arterial stiffness, as supported by prior studies reporting decreased stiffness after isometric forearm contraction [38].

In contrast, the VM has not been shown to elicit significant changes in peripheral PWV (femoral–dorsalis pedis artery), suggesting that baroreceptor adjustments during the VM do not extend to peripheral vascular resistance [39]. Therefore, although the VM may influence central vascular function through baroreceptor regulation, it appears insufficient to recruit limb muscle activity required to induce peripheral vascular changes.

Both interventions produced similar increases in HR and blood pressure. HR rose during both and declined by post2, likely reflecting the shared effect of elevated intrathoracic pressure. In the VM, blood pressure may increase owing to enhanced cardiac output resulting driven by the rise in intrathoracic pressure. In AB, strong abdominal muscle contraction elevates intra-abdominal pressure, pushing the diaphragm upward, reducing lung volume, and thereby increasing intrathoracic pressure and blood pressure [40]. Previous studies suggest that externally applied abdominal pressure can elicit an immediate rise in HR and may play a role attenuating hemodynamic disturbance, including orthostatic [13].

Notably, rSO_2_ in the left prefrontal cortex, measured by near-infrared spectroscopy, decreased significantly after both interventions, whereas HbO peaked immediately afterward and then gradually declined. rSO_2_ reflects prefrontal cortical oxygen saturation, whereas HbO represents oxyhemoglobin levels. Previous research demonstrated a decrease in the total oxygenation index in the prefrontal cortex; in our study, a similar decrease in rSO_2_ was observed, which has been suggested in previous studies to reflect a direct impact on cerebral perfusion due to reduced venous return and increased intracranial pressure associated with elevated intrathoracic pressure [16].

Increases HbO suggests elevated blood flow in response to higher oxygen demand, whereas decreased rSO_2_ indicates that oxygen delivery was insufficient to match demand. The concurrent changes may therefore reflect either heightened metabolic activity or an imbalance between blood flow and oxygen delivery, which limits interpretation [16,41,42,43]. In particular, recovery of rSO_2_ after AB was delayed until post3, indicating slower normalization compared with the VM. This delay is likely attributable to greater oxygen consumption from muscle contractions during AB. In the right hemisphere, rSO_2_ and HbO showed patterns similar to those in the left hemisphere, except for right HbO during the VM, which did not change significantly [44,45].

Therefore, the VM elevated intrathoracic pressure, directly affecting the cardiovascular system and aorta, which increased the PI at the CCA, indicating a primary influence on central blood flow. In contrast, AB raised intra-abdominal pressure through abdominal core muscle contraction while maintaining respiration, leading to greater peripheral muscle activation and oxygen consumption. Consequently, AB demonstrated stronger effects on peripheral blood flow regulation. Clinically, these findings suggest that the VM may be useful for assessing CBF regulation or evoking central vascular responses, whereas AB may provide insight into peripheral vascular responses and support exercise or rehabilitation programs targeting muscle strength and circulatory health.

The present study has some limitations. First, although the VM and AB were performed at the same target pressures, the two interventions differ in how pressure is generated, making it difficult to regard the intervention intensities as identical. Future studies should account for perceived difficulty, for example, through surveys evaluating the effort required for the VM and AB. Second, only a single intervention intensity was tested. Prior studies have shown that cerebral hemodynamic responses to the VM vary with pressures from 20 to 40 mmHg. To enable more precise comparisons, future research should apply AB pressures stepwise and evaluate the hemodynamic effects of both strategies across different pressure ranges. Third, only transient effects were observed. The study captured immediate hemodynamic changes induced by short interventions but did not assess longer-term responses.

Future research should consider comparing various pressure intensities, evaluating the long-term effects of these interventions, and including post-intervention follow-up with repeated interventions at different intensities to clarify sustained or cumulative effects. Although both interventions elevated blood pressure in this study, the VM mainly impacted central vascular function, while AB primarily affected peripheral vasculature. Such differential effects could inform the targeted use of these interventions in exercise programming, rehabilitation, and hemodynamic evaluation. This study is novel in that few previous investigations have directly compared the differential hemodynamic effects of thoracic and abdominal pressure on central and peripheral parameters in healthy participants. It provides a scientifically in-depth perspective on the effects of these interventions on blood flow, complementing prior studies that primarily focused on muscle activation and functional strengthening.

## 5. Conclusions

In this study, the VM showed a significant increase in PI, which caused a transient rise in intrathoracic pressure, directly affecting the cardiovascular system and aorta. In contrast, AB led to decreases in SpO_2_ and peripheral PWV, reflecting an elevation in intra-abdominal pressure through abdominal core muscle contraction, resulting in greater peripheral muscle activation and oxygen consumption. In conclusion, although both interventions increased blood pressure, the VM predominantly influenced the central vascular system, whereas AB primarily affected the peripheral system. This distinction supports their selective application in exercise prescription, rehabilitation strategies, and blood flow assessment.

## Figures and Tables

**Figure 1 medicina-61-02031-f001:**
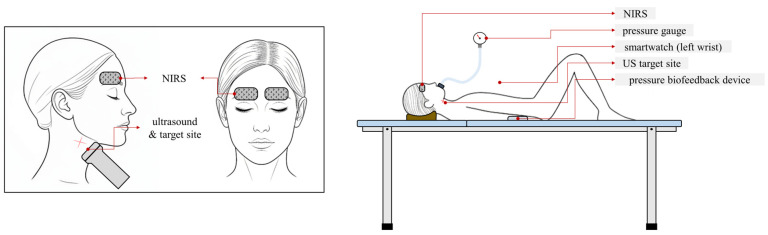
Participant positioning and measurement setup. The participant lay in a hook-lying position with measurement devices attached. These included near-infrared spectroscopy (NIRS) sensors on the forehead, a pressure gauge for monitoring respiratory pressure during the Valsalva maneuver, a smartwatch on the left wrist, an ultrasound (US) probe (B-mode and Doppler) at the carotid site, and a pressure biofeedback device for the abdominal bracing task.

**Figure 2 medicina-61-02031-f002:**
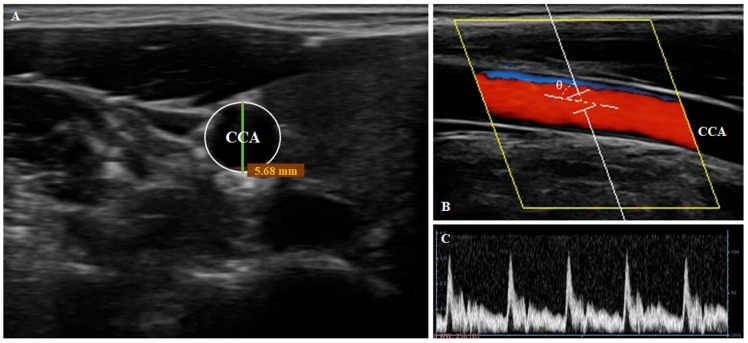
Ultrasound images of the common carotid artery (CCA). (**A**) B-mode image showing the CCA lumen diameter. (**B**) Color Doppler image illustrating the Doppler angle (60°) and sample volume. (**C**) Doppler spectral waveform of the CCA.

**Figure 3 medicina-61-02031-f003:**
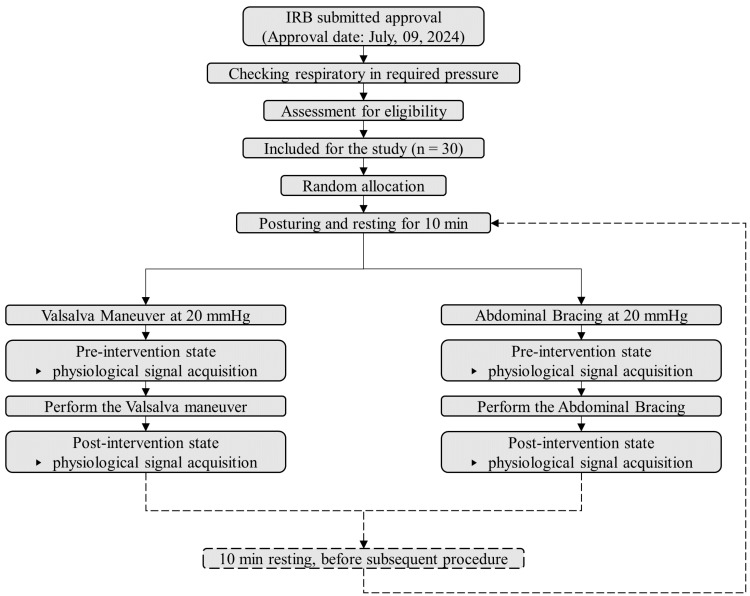
Flowchart of the experimental procedure. Physiological data included peripheral capillary oxygen saturation, heart rate, blood pressure, minimum and maximum vessel diameters, pulsatility index, resistive index, and heart–finger pulse wave velocity. All parameters were measured before and after each task. IRB, institutional review board.

**Figure 4 medicina-61-02031-f004:**
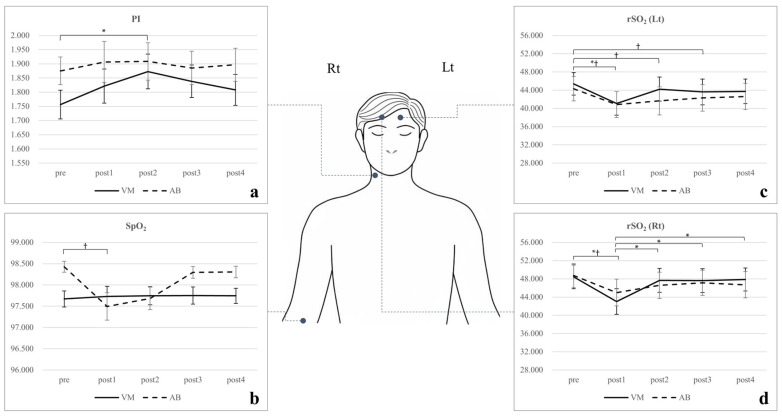
Interval-based changes in hemodynamic parameters during the Valsalva maneuver (VM) and abdominal bracing (AB). Data are presented as mean ± standard error of the mean. (**a**) Pulsatility index (PI). (**b**) Peripheral capillary oxygen saturation (SpO_2_). (**c**) Regional oxygen saturation (rSO_2_) in the left (Lt) prefrontal cortex. (**d**) rSO_2_ in the right (Rt) prefrontal cortex. * *p* < 0.05 for VM; † *p* < 0.05 for AB; *† *p* < 0.05 for both.

**Table 1 medicina-61-02031-t001:** Simplified comparison of hemodynamic responses (pulsatility index, regional oxygen saturation and peripheral capillary oxygen saturation) during the Valsalva maneuver and abdominal bracing.

		Repeated-Measures Comparison					Post Hoc Comparison (Tukey)		
Group	Parameter	Variable	Mean ± SD	F	*p*	Repeated-Measures Factors		t	*p* _Tukey_
VM	PI	Pre	1.76 ± 0.28	4.48	0.008 *	Pre	Post2	−3.29	0.020 *
		Post1	1.82 ± 0.33						
		Post2	1.87 ± 0.33						
		Post3	1.84 ± 0.32						
		Post4	1.81 ± 0.30						
	rSO_2_ (Lt)	Pre	45.37 ± 13.09	5.70	0.005 *	Pre	Post1	3.26	0.023 *
		Post1	41.12 ± 13.64						
		Post2	44.24 ± 14.03						
		Post3	43.61 ± 14.90						
		Post4	43.74 ± 14.27						
	rSO_2_ (Rt)	Pre	48.46 ± 13.78	10.99	<0.001 *	Pre	Post1	4.06	0.003 *
		Post1	43.02 ± 14.91						
		Post2	47.68 ± 13.88			Post1	Post2	−3.23	0.025 *
		Post3	47.63 ± 13.82				Post3	−3.40	0.016 *
		Post4	47.86 ± 13.38				Post4	−3.53	0.012 *
AB	SpO_2_	Pre	98.43 ± 0.71	5.56	<0.001 *	Pre	Post1	3.17	0.027 *
		Post1	97.49 ± 1.76						
		Post2	97.68 ± 1.45						
		Post3	98.30 ± 0.75						
		Post4	98.30 ± 0.75						
	rSO_2_ (Lt)	Pre	44.32 ± 14.03	6.15	0.001 *	Pre	Post1	3.47	0.014 *
		Post1	40.83 ± 15.09						
		Post2	41.66 ± 16.26				Post2	3.08	0.035 *
		Post3	42.30 ± 15.39				Post3	3.32	0.020 *
		Post4	42.60 ± 15.09						
	rSO_2_ (Rt)	Pre	48.74 ± 13.95	5.95	0.002 *	Pre	Post1	3.36	0.018 *
		Post1	45.00 ± 15.48						
		Post2	46.58 ± 15.21						
		Post3	47.13 ± 14.62						
		Post4	46.70 ± 15.32						

Abbreviations: VM, Valsalva maneuver; AB, abdominal bracing; PI, pulsatility index; rSO_2_, regional oxygen saturation; SpO_2_, peripheral capillary oxygen saturation; Lt, left; Rt, right; SD, standard deviation; * *p* < 0.05.

## Data Availability

The original contributions presented in this study are included in the published article material. Additional raw data underlying this study available from the corresponding author upon reasonable request, in accordance with participant privacy regulations.

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
