# Peer review of "Comparative Effects of Abdominal Bracing and Valsalva Maneuver on Cerebral and Peripheral Hemodynamics in Healthy Adults: A Randomized Crossover Study"

_medicina, 2025, doi:10.3390/medicina61112031_

Round 1

Reviewer 1 Report

Comments and Suggestions for Authors

Dear authors,
I appreciate the opportunity to review your manuscript, "Comparative Effects of the Abdominal Brace and the Valsalva Maneuver on Cerebral and Peripheral Hemodynamics." I believe it may contribute something new to the scientific community due to its comparison of the "abdominal brace" intervention versus the Valsalva maneuver, measuring the differential hemodynamic effects between cerebral and peripheral circulation with the variables of cerebral blood flow, peripheral arterial stiffness, and peripheral and cerebral oxygen saturation, contrasting central versus peripheral vascular mechanisms as you propose.
Please allow me to make a few comments:
- Title: It is clear and direct, reflecting comparative interventions. I suggest adding the study subjects (healthy subjects) and the research design (Experimental repeated measures trial with a randomized crossover design in healthy subjects?).
- Abstract: I would add the study design under methods.
- Introduction: I declare that I have no conflicts of interest in any of the references I recommend below. I believe they could provide recent evidence on the systemic and cerebrovascular hemodynamic effects that your current references do not address and justify the use of specific technologies (such as NIRS) and comparison with the Valsalva maneuver:
-Jung et al., 2024 – Cerebrovascular effects of MV at different pressures.
-Rabie et al., 2021 – NIRS and intra-abdominal pressure: validity of cerebral oxygenation.
-Mitra et al., 2025 – Effects of abdominal compression on hemodynamics in healthy subjects.
-Nakai et al., 2023 – Intervention with abdominal support and its functional effects.
-Pstras et al., 2016 – Complete physiology of the Valsalva maneuver and its phases.
I would also highlight the originality of their research, given the lack of studies directly comparing the differential hemodynamic effects of intrathoracic (MV) versus intra-abdominal (corset) pressure on central and peripheral parameters in healthy subjects.
- Materials and methods: It is recommended to follow and complete the STROBE checklist for observational studies with a controlled intervention.
- Add a study design section and specify if there was a washout period between the two interventions.
- For participants, I suggest clearly adding participant inclusion criteria.
- Experimental procedure: Describe how the interventions were randomized.
- Results:
- Add a description of the sample (even if you added it in the methods section).
- The authors clearly present a large amount of data, which provides valuable information. I recommend adding a single summary table that compares the main findings of both interventions (MV vs. AB), clearly highlighting which variables were most affected by each. - Discussion: Relate your results to the existing literature, indicating the author: Do they agree or differ? Why?
- Conclusions: I believe that if you left only lines 395 to 399, they would clearly summarize your findings and their clinical implications (the rest is methodology and discussion).
Thank you very much.

Author Response

Authors’ Responses to Reviewers’ Comments:

We sincerely appreciate your valuable comments and suggestions, which helped us improve the quality of our manuscript. We carefully revised the manuscript following the reviewer’s recommendations accordingly. Point-by-point responses are as follows:

Comments and Suggestions for Authors:

-----------------------------------------------------------------------------------------------------

Reviewer #1:

Dear authors,

I appreciate the opportunity to review your manuscript, "Comparative Effects of the Abdominal Brace and the Valsalva Maneuver on Cerebral and Peripheral Hemodynamics." I believe it may contribute something new to the scientific community due to its comparison of the "abdominal brace" intervention versus the Valsalva maneuver, measuring the differential hemodynamic effects between cerebral and peripheral circulation with the variables of cerebral blood flow, peripheral arterial stiffness, and peripheral and cerebral oxygen saturation, contrasting central versus peripheral vascular mechanisms as you propose.

Please allow me to make a few comments:

Specific comments:

  1. Title: It is clear and direct, reflecting comparative interventions. I suggest adding the study subjects (healthy subjects) and the research design (Experimental repeated measures trial with a randomized crossover design in healthy subjects?).

Response: Thank you for your valuable suggestion. Following the reviewer's recommendation, we have revised the title to include information on the study design as follows: ‘Comparative Effects of Abdominal Bracing and the Valsalva Maneuver on Cerebral and Peripheral Hemodynamics in Healthy Adults: A Randomized Crossover Study.’

> Title section (pg. 1, line 2): “Comparative Effects of Abdominal Bracing and the Valsalva Maneuver on Cerebral and Peripheral Hemodynamics in Healthy Adults: A Randomized Crossover Study”

  1. Abstract: I would add the study design under methods.

Response: We appreciate your insightful comment. Following the reviewer's suggestion, we have added information on the study design of this experimental study—a randomized crossover design—and the statistical methods (repeated-measures analysis of variance and paired t-tests) to the Materials and Methods section of the Abstract.

> Abstract section (pg. 1, line 27):Materials and Methods: A randomized crossover design was used and thirty healthy young adults (age 21.9 ± 1.5 years; BMI 20.9 ± 1.8 kg/m²) performed AB and VM in randomized order. All participants provided written informed consent, and the study protocol was approved by the Clinical Research Information Service (KCT0009742; registered on 30 August 2024). Hemodynamic responses were measured before and after each intervention, including heart rate, blood pressure, pulse wave velocity, carotid artery diameter, pulsatility index, resistive index, peripheral oxygen saturation, and cerebral oxygenation. Repeated-measures analysis of variance and paired t-tests were conducted on the datasets.”

  1. Introduction: I declare that I have no conflicts of interest in any of the references I recommend below. I believe they could provide recent evidence on the systemic and cerebrovascular hemodynamic effects that your current references do not address and justify the use of specific technologies (such as NIRS) and comparison with the Valsalva maneuver:
    - Jung et al., 2024 – Cerebrovascular effects of MV at different pressures.
    - Rabie et al., 2021 – NIRS and intra-abdominal pressure: validity of cerebral oxygenation.
    - Mitra et al., 2025 – Effects of abdominal compression on hemodynamics in healthy subjects.
    - Nakai et al., 2023 – Intervention with abdominal support and its functional effects.
    - Pstras et al., 2016 – Complete physiology of the Valsalva maneuver and its phases.

I would also highlight the originality of their research, given the lack of studies directly comparing the differential hemodynamic effects of intrathoracic (MV) versus intra-abdominal (corset) pressure on central and peripheral parameters in healthy subjects.

Response: Thank you for your valuable suggestions. We have additionally cited the paper suggested by the reviewer, Mitra et al., 2025. Based on the reference, we mention in the Discussion that previous studies reported an immediate increase in heart rate following externally applied abdominal pressure, and that abdominal pressure may help mitigate hemodynamic changes such as orthostatic hypotension. In contrast, Jung et al., 2024 reports findings from the author’s own research group, and other references were not directly relevant; therefore, they were not additionally cited.

Furthermore, following the reviewer's suggestion, we have added a sentence at the end of the Discussion emphasizing the originality of our study, noting that few studies have directly compared the differential hemodynamic effects of intrathoracic and intra-abdominal pressure on central and peripheral parameters in healthy participants.

> Discussion section (pg. 11, line 387): “Previous studies suggest that externally applied abdominal pressure can elicit an immediate rise in HR and may play a role attenuating hemodynamic disturbance, including orthostatic [13].”

> Discussion section (pg. 12, line 434): “This study is novel in that few previous investigations have directly compared the differential hemodynamic effects of thoracic and abdominal pressure on central and peripheral parameters in healthy participants.”

  1. Materials and methods: It is recommended to follow and complete the STROBE checklist for observational studies with a controlled intervention.

Response: Thank you for pointing this out. For this study, we completed and submitted the TREND checklist, which is the clinical trial checklist required by Medicina.

  1. Add a study design section and specify if there was a washout period between the two interventions.

Response: We appreciate your insightful comment. Although the washout period between interventions was mentioned in the main text under “Experimental Protocol,” we have added explicit wording to clarify the washout period. A 10-minute washout period was given between the two interventions.

> Materials and Methods section (pg. 3, line 110):
“2.1. Study Design

This study was conducted as a randomized crossover experimental study with reference to the Transparent Reporting of Evaluations with Nonrandomized Designs (TREND) checklist. The study was conducted in accordance with the Declaration of Helsinki, and approved by the Institutional Review Board (approval date: 09 July 2024; approval number: 1044396-202405-HR-088-01) and registered in the Clinical Research Information System (date of registration: 30 August 2024; registration number: KCT0009742; patient enrollment and trial start date: 01 September 2024; trial end date: 31 December 2024). All participants provided written informed consent before beginning the study.”

> Materials and Methods section (pg. 6, line 236): “A washout period of 10-minute was provided between the two interventions.”

  1. For participants, I suggest clearly adding participant inclusion criteria.

Response: In this study, participants were recruited who had no cerebrovascular, neurological, or other medical conditions. Exclusion criteria included: conditions affecting cerebral blood flow; abnormal neurological findings; diseases or medications related to impaired brain function; respiratory disorders; and participants with abnormal ranges of BMI, blood pressure, or other indicators that could affect cardiovascular function. We will describe this section more clearly.

> Materials and Methods section (pg. 3, line 120): “Thirty healthy adults in their twenties were included, meeting the inclusion criteria of having no cerebrovascular, neurological, or other medical conditions. The exclusion criteria included a history of diseases affecting CBF, abnormal neurological findings, disorders or medications related to cerebral dysfunction, or respiratory conditions.”

  1. Experimental procedure: Describe how the interventions were randomized.

Response: We appreciate your comment. The order of assignment was randomized using the Excel random function, with 15 participants performing AB followed by VM, and 15 participants performing VM followed by AB. We will add this information accordingly.

> Materials and Methods section (pg. 6, line 234): “The order of assignment was randomized using the Excel random function, with 15 participants performing AB followed by VM, and 15 participants performing VM followed by AB.”

  1. Results: Add a description of the sample (even if you added it in the methods section).

Response: Following the reviewer's suggestion, we have added a sentence regarding the sample at the beginning of the Results section.

> Results section (pg. 8, line 267): “This study was conducted with thirty healthy participants. All participants were ethnically homogeneous and were university students. No participants withdrew from the experiment, and data from all thirty participants were included in the analysis.”

  1. The authors clearly present a large amount of data, which provides valuable information. I recommend adding a single summary table that compares the main findings of both interventions (MV vs. AB), clearly highlighting which variables were most affected by each.

Response: Thank you for your constructive suggestion. Although the main outcomes are presented as tables, we will add a simplified table showing the numerical data for results that demonstrate statistical significance to support the graphical presentation (e.g., PI, SpO2, rSO2).

> Results section (pg. 9, line 322):

Table 1. Simplified comparison of hemodynamic responses (pulsatility index, regional oxygen saturation and peripheral capillary oxygen saturation) during the Valsalva maneuver and abdominal bracing.

Repeated-measures comparison

Post hoc comparison (Tukey)

Group

Parameter

Variable

Mean ± SD

F

p-value

Repeated-measures factor

T

pTukey

VM

PI

Pre

1.76 ± 0.28

4.48

0.008*

Pre

Post2

–3.29

0.020*

Post1

1.82 ± 0.33

Post2

1.87 ± 0.33

Post3

1.84 ± 0.32

Post4

1.81 ± 0.30

rSO2 (Lt)

Pre

45.37 ± 13.09

5.70

0.005*

Pre

Post1

3.26

0.023*

Post1

41.12 ± 13.64

Post2

44.24 ± 14.03

Post3

43.61 ± 14.90

Post4

43.74 ± 14.27

rSO2 (Rt)

Pre

48.46 ± 13.78

10.99

<0.001*

Pre

Post1

4.06

0.003*

Post1

43.02 ± 14.91

Post2

47.68 ± 13.88

Post1

Post2

–3.23

0.025*

Post3

47.63 ± 13.82

Post3

–3.40

0.016*

Post4

47.86 ± 13.38

Post4

–3.53

0.012*

AB

SpO2

Pre

98.43 ± 0.71

5.56

<0.001*

Pre

Post1

3.17

0.027*

Post1

97.49 ± 1.76

Post2

97.68 ± 1.45

Post3

98.30 ± 0.75

Post4

98.30 ± 0.75

rSO2 (Lt)

Pre

44.32 ± 14.03

6.15

0.001*

Pre

Post1

3.47

0.014*

Post1

40.83 ± 15.09

Post2

41.66 ± 16.26

Post2

3.08

0.035*

Post3

42.30 ± 15.39

Post3

3.32

0.020*

Post4

42.60 ± 15.09

rSO2 (Rt)

Pre

48.74 ± 13.95

5.95

0.002*

Pre

Post1

3.36

0.018*

Post1

45.00 ± 15.48

Post2

46.58 ± 15.21

Post3

47.13 ± 14.62

Post4

46.70 ± 15.32

Abbreviations: VM, Valsalva maneuver; AB, abdominal bracing; PI, pulsatility index; rSO2, regional oxygen saturation; SpO2, peripheral capillary oxygen saturation; Lt, left; Rt, right; SD, standard deviation; * p < 0.05.

  1. Discussion: Relate your results to the existing literature, indicating the author: Do they agree or differ? Why?

Response: Following the reviewer’s comments, we have added the following sentences. First, previous studies have reported that heart rate increases immediately after abdominal pressure and that abdominal pressure may help mitigate hemodynamic changes such as orthostatic hypotension added as reference [added as reference 13]. Second, previous research demonstrated a decrease in the total oxygenation index in the prefrontal cortex; in our study, a similar decrease in rSOâ‚‚ was observed, which has been suggested in previous studies to reflect a direct impact on cerebral perfusion due to reduced venous return and increased intracranial pressure associated with elevated intrathoracic pressure [10].

[13] Mitra, K.; Kunte, S. A.; Taube, S. E.; Sankarlinkam, S.; Mohamed, L.; Adodo, E.; Green, C. L.; Fudim, M.; Richardson, E. S. Standing under Pressure: Hemodynamic Effects of Abdominal Compression Type and Intensity in Healthy Adults. Front. Physiol. 2025, 16. https://doi.org/10.3389/fphys.2025.1621617.

[10] Perry, B. G.; Cotter, J. D.; Mejuto, G.; Mündel, T.; Lucas, S. J. E. Cerebral Hemodynamics during Graded Valsalva Maneuvers. Front Physiol 2014, 5, 349. https://doi.org/10.3389/fphys.2014.00349.

> Discussion section (pg. 11, line 387): “Previous studies suggest that externally applied abdominal pressure can elicit an immediate rise in HR and may play a role attenuating hemodynamic disturbance, including orthostatic [13].”

> Discussion section (pg. 11, line 394): “Previous research demonstrated a decrease in the total oxygenation index in the prefrontal cortex; in our study, a similar decrease in rSOâ‚‚ was observed, which has been suggested in previous studies to reflect a direct impact on cerebral perfusion due to reduced venous return and increased intracranial pressure associated with elevated intrathoracic pressure [16].”

  1. Conclusions: I believe that if you left only lines 395 to 399, they would clearly summarize your findings and their clinical implications (the rest is methodology and discussion).

Response: Based on the reviewer’s comments, we have revised the Conclusion to clearly summarize the study findings and their clinical implications as follows.

> Discussion section (pg. 12, line 408): “Therefore, the VM elevated intrathoracic pressure, directly affecting the cardiovascular system and aorta, which increased the PI at the CCA, indicating a primary influence on central blood flow. In contrast, AB raised intra-abdominal pressure through abdominal core muscle contraction while maintaining respiration, leading to greater peripheral muscle activation and oxygen consumption. Consequently, AB demonstrated stronger effects on peripheral blood flow regulation.

Clinically, these findings suggest that the VM may be useful for assessing CBF regulation or evoking central vascular responses, whereas AB may provide insight into peripheral vascular responses and support exercise or rehabilitation programs targeting muscle strength and circulatory health.”

> Conclusion section (pg. 12, line 443): “In conclusion, although both interventions increased blood pressure, the VM predominantly influenced the central vascular system, whereas AB primarily affected the peripheral system. This distinction supports their selective application in exercise prescription, rehabilitation strategies, and blood flow assessment. Future studies should compare different pressure intensities and evaluate long-term effects to expand on these findings.”

Reviewer 2 Report

Comments and Suggestions for Authors

Dear Authors,

I want to express my gratitude for the opportunity to revise this manuscript.

The article addresses a very pertinent topic, congratulations. Below are specific suggestions aiming the manuscript improvement, with line indication.:

The major issues are:

51-98 - Please consider standardizing the paragraphs´ size to improve readability. 295-384 – The same should be considered for the discussion section.

100-120 - Please describe all available information related to the subjects´ characterization. Some examples, training routines, years of experience, competitive level, number of weekly training sessions (specific/strength/injury prevention), and games/competitive events. Another important piece of information is, for example, whether subjects refrain from intense exercise before data collection. Please describe the ethical details. Informed consents fulfilled?

121-215 - Please describe all methodological details, for example, the equipment (manufacturer, version, city, and country), as well as the associated procedures in detail (including time of the season, familiarization, environmental conditions, nutrition, equipment, medicine, warm-up, human resources involved – academic background and experience), preferably with reference support.

291 - Please revise all tables´ content and format. For example, in table 5, the Mean ± SD  values are not in the same line.

384 - Please consider providing suggestions for future research.

- - -

The minor issues:

28-36 / 36-42 - Please consider reducing the methodological information and providing more numerical results.

179 - Subtitle in uppercase, in other cases in lowercase. Please standardize the format.

198 - NIRS abbreviated / 217 – In full. Please revise all abbreviations throughout the manuscript.

216 - Please include all information. For example, the presentation of data by mean and SD.

226 - Please consider the “p” in italics.

431 - Please double-check the references format, considering the journal template. For example, the ref 28 format of the journal is different compared with others.

Please consider improving the quality of the figures.

Please revise the document format considering the journal template.

Please revise the English details throughout the manuscript.

Comments on the Quality of English Language

The English is globally with good quality. Some details can be improved.

Author Response

Authors’ Responses to Reviewers’ Comments:

We sincerely appreciate your valuable comments and suggestions, which helped us improve the quality of our manuscript. We carefully revised the manuscript following the reviewer’s recommendations accordingly. Point-by-point responses are as follows:

Comments and Suggestions for Authors:

-----------------------------------------------------------------------------------------------------

Reviewer #2:

Dear Authors,

I want to express my gratitude for the opportunity to revise this manuscript.

The article addresses a very pertinent topic, congratulations. Below are specific suggestions aiming the manuscript improvement, with line indication.:

Specific comments:

The major issues are:
1. 51-98 - Please consider standardizing the paragraphs´ size to improve readability.

Response: Thank you for your comment. The structure of the main text is organized as follows: the importance of blood circulation, key indicators of blood circulation, interventions to induce changes in the primary indicator, blood pressure, and the rationale for the study. Following the reviewer’s suggestion, we have adjusted the paragraph lengths in the Introduction to standardize them and improve readability.

  1. 295-384 – The same should be considered for the discussion section.

Response: Following the reviewer’s suggestion, we have also adjusted the paragraph lengths in the Discussion to improve readability.

  1. 100-120 - Please describe all available information related to the subjects´ characterization. Some examples, training routines, years of experience, competitive level, number of weekly training sessions (specific/strength/injury prevention), and games/competitive events. Another important piece of information is, for example, whether subjects refrain from intense exercise before data collection. Please describe the ethical details. Informed consents fulfilled?

Response: We appreciate your generous comments. Participants in this study were healthy adults in their 20s. Individuals who had received professional training or education related to exercise, or who engaged in regular physical activity (such as strength training, aerobic exercise, or conditioning) were excluded. Therefore, most participants had little to no exercise experience or engaged in irregular physical activity. Participants were instructed to avoid any stimuli that could induce hemodynamic changes for 24 hours prior to participation, and they took part in the experiments in a rested state. In addition, all participants provided written informed consent, and we have included this information in the manuscript.

> Materials and Methods section (pg. 3, line 123): “In addition, individuals with professional training or education related to exercise, or those engaged in high-intensity regular physical activity (such as strength training, aerobic exercise, or conditioning) were excluded. To minimize confounding, participants were instructed to avoid substances affecting the vascular or autonomic nervous systems (caffeine, alcohol, and smoking) for 24 h before testing and they took part in the experiments in a rest state.”

  1. 121-215 - Please describe all methodological details, for example, the equipment (manufacturer, version, city, and country), as well as the associated procedures in detail (including time of the season, familiarization, environmental conditions, nutrition, equipment, medicine, warm-up, human resources involved – academic background and experience), preferably with reference support.

Response: Thank you for your constructive feedback. All ultrasound examinations were performed by an operator with more than four years of ultrasound research experience (co-corresponding author, Dr. Ju-Yeon Jung). Several ultrasound-related studies have been published by Dr. Jung since 2022 [refs. 22, 23, which have been added to complement ref. 15]. Additional detailed descriptions of the ultrasound protocol and equipment have now been provided as follows.

Ultrasound measurements were acquired using a linear transducer (L3-12T, 3–12 MHz) with the following preset: depth of 4.0 cm, dynamic range of 42 dB, gain of 36, wall filter of 2, and pulse repetition frequency (PRF) of 3.6 kHz. Color Doppler was used to confirm the direction of blood flow. A 1.0-mm sample volume was placed at the center of the long-axis carotid artery, and the angle-correction cursor was aligned parallel to the vessel centerline. The Doppler angle was fixed at 60° to ensure reproducibility. The measurement site was marked for repeated assessment at the same location. Doppler parameters, including PSV and PI, were recorded every second for 60 seconds, resulting in 60 data points.

Carotid artery diameters were measured using B-mode imaging. ECG gating was not applied; instead, systole and diastole were determined based on the maximum and minimum luminal diameters, respectively. A 5-second cine loop was recorded to verify systolic and diastolic changes. The intima–intima boundary tracking method was applied [added as reference 24]. Among the 170 frames recorded over 5 seconds, the smallest and largest diameters were defined as Dmin and Dmax, respectively.

In addition, a description was added regarding the STD-1000 system (StraTek Co., Korea), in which blood flow transit time is calculated from the delay between ECG and finger PPG signals, and the path length is estimated based on the participant’s height (height × 0.56).

Relevant and newly added references:

[22] Jung, J.-Y.; Cho, H.-Y.; Kang, C.-K. Effects of a Traction Device for Head Weight Reduction and Neutral Alignment during Sedentary Visual Display Terminal (VDT) Work on Postural Alignment, Muscle Properties, Hemodynamics, Preference, and Working Memory Performance. International Journal of Environmental Research and Public Health 2022, 19 (21), 14254. https://doi.org/10.3390/ijerph192114254.

[23] Yang, E.-S.; Jung, J.-Y.; Kang, C.-K. Effects of Low-Pressure Valsalva Maneuver on Changes in Cerebral Arterial Stiffness and Pulse Wave Velocity. PLOS ONE 2024, 19 (9), e0308866. https://doi.org/10.1371/journal.pone.0308866.

[15] Jung, J.-Y.; Lee, Y.-B.; Kang, C.-K. Effect of Controlled Expiratory Pressures on Cerebrovascular Changes During Valsalva Maneuver. Applied Sciences 2024, 14 (22), 10132. https://doi.org/10.3390/app142210132.

[24] Stolz, L. A.; Mosier, J. M.; Gross, A. M.; Douglas, M. J.; Blavais, M.; Adhikari, S. Can Emergency Physicians Perform Common Carotid Doppler Flow Measurements to Assess Volume Responsiveness? Western Journal of Emergency Medicine: Integrating Emergency Care with Population Health 2015, 16 (2). https://doi.org/10.5811/westjem.2015.1.24301.

> Materials and Methods section (pg. 4, line 158): “The preset of ultrasound measurements is as follows: a depth of 4.0 cm, dynamic range of 42 dB, gain of 36, wall filter of 2, and pulse repetition frequency (PRF) of 3.6 kHz. The CCA was chosen for its accessibility and established relevance in cerebral hemodynamic studies [20,21]. Furthermore, all ultrasound examinations were performed by an operator with more than four years of ultrasound research experience [15,22,23].”

> Materials and Methods section (pg. 4, line 167): “The minimum (Dmin) and maximum (Dmax) lumen diameters were measured from these images using the intima-intima boundary tracking method in RadiAnt DICOM Viewer (version 2025.1; Medixant, 2025; https://www.radiantviewer.com; Figure 1a) [24].”

> Materials and Methods section (pg. 5, line 184): “To minimize error and improve accuracy, color Doppler was used to confirm the direction of blood flow, and the Doppler angle was fixed at 60° (Figure 1b, 1c). Additionally, a 1.0 mm sample volume was placed at the center of the long-axis carotid artery, and the angle correction cursor was aligned parallel to the vessel centerline.”

> Materials and Methods section (pg. 5, line 193): “The method is based on pulse transit time and subject height (used to estimate arm length; height × 0.56).”

  1. 291 - Please revise all tables´ content and format. For example, in table 5, the Mean ± SD values are not in the same line.

Response: We have revised all tables (Table S1-S5) as the suggested by the reviewer.

> Supplementary Materials section (pg. 17, line 597): Table S1. Comparison of heart rate and peripheral capillary oxygen saturation responses be-tween the Valsalva maneuver and abdominal bracing.

> Supplementary Materials section (pg. 18, line 601): Table S2. Comparison of blood pressure responses (systolic and diastolic) during the Valsalva maneuver and abdominal bracing.

> Supplementary Materials section (pg. 19, line 606): Table S3. Comparison of vascular hemodynamic responses (pulsatility index and resistive index) during the Valsalva maneuver and abdominal bracing.

> Supplementary Materials section (pg. 20, line 611): Table S4. Comparison of vascular hemodynamic responses (carotid pulse wave velocity, vessel diameters, heart–finger pulse wave velocity) during the Valsalva maneuver and abdominal bracing.

> Supplementary Materials section (pg. 21, line 617): Table S5. Comparison of cerebral hemodynamic responses (regional oxygen saturation and oxyhemoglobin) during the Valsalva maneuver and abdominal bracing.

  1. 384 - Please consider providing suggestions for future research.

Response: We appreciate your valuable comment. In response to the reviewer’s suggestion, we have added a section in the Discussion highlighting the originality of our study and suggesting directions for future research.

> Discussion section (pg. 12, line 439): “Future research should consider comparing various pressure intensities and evaluating the long-term effects of these interventions to further extend these findings.”

The minor issues:

  1. 28-36 / 36-42 - Please consider reducing the methodological information and providing more numerical results.

Response: Thank you for your valuable suggestion. In the Materials and Methods section of the Abstract, we omitted mention of the measurement devices, focused on the key outcome measures, and added the mean ± SD values and p-values for each intervention to clarify the results for each outcome.

> Abstract section (pg. 1, line 27): Materials and Methods: A randomized crossover design was used and thirty healthy young adults (age 21.9 ± 1.5 years; BMI 20.9 ± 1.8 kg/m²) performed AB and VM in randomized order. All participants provided written informed consent, and the study protocol was approved by the Clinical Research Information Service (KCT0009742; registered on 30 August 2024). Hemodynamic responses were measured before and after each intervention, including heart rate, blood pressure, pulse wave velocity, carotid artery diameter, pulsatility index, resistive index, peripheral oxygen saturation, and cerebral oxygenation. Repeated-measures analysis of variance and paired t-tests were conducted on the datasets.”

> Abstract section (pg. 1, line 35): Results: Both the VM and AB significantly increased heart rate (p < 0.001), and systolic blood pressure (VM: p = 0.015; AB: p < 0.001). Cerebral oxygen saturation decreased significantly (VM: p < 0.05; AB: p < 0.05), whereas oxyhemoglobin increased during both interventions, suggesting higher cerebral oxygen demand. The VM specifically increased the carotid pulsatility index (pre = 1.76 ± 0.28; post2 = 1.87 ± 0.33; p = 0.008), reflecting elevated central vascular resistance. In contrast, AB decreased peripheral oxygen saturation (pre = 98.43 ± 0.71; post1 = 97.49 ± 1.76; p < 0.001) and increased peripheral (heart–finger) pulse wave velocity (Lt: p = 0.026; Rt: p = 0.010), indicating greater stimulation of peripheral circulation.”

  1. 179 - Subtitle in uppercase, in other cases in lowercase. Please standardize the format.

Response: We have made the requested corrections to standardize the subtitle format.

  1. 198 - NIRS abbreviated / 217 – In full. Please revise all abbreviations throughout the manuscript.

Response: Following the reviewer’s advice, we have used abbreviations consistently after their first mention.

  1. 216 - Please include all information. For example, the presentation of data by mean and SD.

Response: Following the reviewer’s suggestion, we have added additional information on the presentation of the data in the Materials and Methods section.

> Materials and Methods section (pg. 7, line 254): “The results were presented as the mean ± standard deviation (SD), F-values, and p-value for each variable, along with t-value, Tukey-adjusted p-value for each factor, and effect size.”

  1. 226 - Please consider the “p” in italics.

Response: We have changed all p-values, including those in tables, to italics (p).

  1. 431 - Please double-check the references format, considering the journal template. For example, the ref 28 format of the journal is different compared with others.

Response: Following the reviewer’s suggestion, we have corrected typographical errors in the references. Although the format of reference 28 appeared consistent with the other citations (author, title, journal, year, volume(issue), page numbers, doi), we have revised it once more for confirmation.

  1. Please consider improving the quality of the figures.

Please revise the document format considering the journal template.

Please revise the English details throughout the manuscript.

Response: Following the reviewer’s suggestions, we have revised the figures to improve their resolution and overall visual quality. We have also adjusted the document formatting to align with the journal’s template requirements. In addition, we have thoroughly reviewed and refined the English throughout the manuscript to enhance clarity and readability.

Reviewer 3 Report

Comments and Suggestions for Authors

The manuscript entitled “Comparative Effects of Abdominal Bracing and Valsalva Maneuver on Cerebral and Peripheral Hemodynamics” was evaluated. The manuscript presents important clinical questions that should be developed and applied with patients. However, for the work to be worthy of publication, some adjustments must be made.

Below is the information that should be considered by authors. I kindly ask that all changes made to the manuscript be highlighted in a different color so that reviewers can find the adjusted items.

Abstract:
1- Please try to keep the study objectives as close as possible to the title. You can even adjust the title to include the main statistical analysis performed and the type of study.
2- Begin the methodology by presenting the type of study.
3- Describe the anthropometric characteristics of the sample, as well as body composition, in the methodology. Also present the statistical analyses that will be performed.
4- In the results, in addition to presenting p-values, please include the means and standard deviation.
5- In conclusion, focus on concluding only the proposed objectives.

Keywords:
6- Avoid repeating words already mentioned in the title, please replace them with synonyms.

Introduction:
7- Begin the introduction with epidemiological information on the topic, then begin presenting the study questions more specifically.
8- Clearly present the study hypotheses, as well as the current state of the art and how this work will contribute to the evolution of the field.
9- At the end of the introduction, please adjust the objectives in the same way as those presented in the abstract.

Materials and Methods:
10- Begin writing the methodology by presenting the type of study and the location where the intervention was performed.
11- Present the technical information of the equipment used for the anthropometric assessment of the participants.
12- Was the study registered as a clinical trial?
13- Please describe the statistical analyses in chronological order, including the normalization tests, how the data were presented, and which tests were used.
14- Check the quality of the flowchart resolution; it is blurry.

Results:
15- Begin reporting the results with the sociodemographic information of the study sample.
16- Regarding t-tests, please also include information on the effect size. Include this information in statistical analyses, tables, and/or in the body of the text.

Discussion:
17- It's not clear to me why you're presenting figures in the discussion. Shouldn't this information be included in the results? Check and adjust.
18- At the end of the discussion, after addressing the limitations, please include the practical applications of the research.
19- Also, try to clarify in the discussion how this research will influence the state of the art on the topic.

Conclusions:
20- The conclusion is too long; please shorten it to address only the proposed objectives.

Regarding the bibliographic references, I found that many are more than 5 years old. Therefore, I suggest you replace older references with more recent ones, that is, from 2022 onward.

Author Response

Authors’ Responses to Reviewers’ Comments:

We sincerely appreciate your valuable comments and suggestions, which helped us improve the quality of our manuscript. We carefully revised the manuscript following the reviewer’s recommendations accordingly. Point-by-point responses are as follows:

Comments and Suggestions for Authors:

-----------------------------------------------------------------------------------------------------

Reviewer #3:

The manuscript entitled “Comparative Effects of Abdominal Bracing and Valsalva Maneuver on Cerebral and Peripheral Hemodynamics” was evaluated. The manuscript presents important clinical questions that should be developed and applied with patients. However, for the work to be worthy of publication, some adjustments must be made.

Below is the information that should be considered by authors. I kindly ask that all changes made to the manuscript be highlighted in a different color so that reviewers can find the adjusted items.

Response: Instead of highlighting, we have added notes and comment numbers to clearly indicate the revised parts in the revised manuscript.

Specific comments:

Abstract:

  1. Please try to keep the study objectives as close as possible to the title. You can even adjust the title to include the main statistical analysis performed and the type of study.

Response: Thank you for your valuable suggestion. In line with the comment from the previous reviewer, following the reviewer's recommendation, we have revised the title to include information on the study design as follows: ‘Comparative Effects of Abdominal Bracing and the Valsalva Maneuver on Cerebral and Peripheral Hemodynamics in Healthy Adults: A Randomized Crossover Study.’

> Title section (pg. 1, line 2): “Comparative Effects of Abdominal Bracing and the Valsalva Maneuver on Cerebral and Peripheral Hemodynamics in Healthy Adults: A Randomized Crossover Study”

  1. Begin the methodology by presenting the type of study.

Response: We appreciate this valuable comment. As this study follows a crossover design, we have added this information at the beginning of the Materials and Methods section in the Abstract, as recommended by the reviewer.

> Abstract section (pg. 1, line 27):Materials and Methods: A randomized crossover design was used and thirty healthy young adults (age 21.9 ± 1.5 years; BMI 20.9 ± 1.8 kg/m²) performed AB and VM in randomized order.”

  1. Describe the anthropometric characteristics of the sample, as well as body composition, in the methodology. Also present the statistical analyses that will be performed.

Response: Following the reviewer’s suggestion, we have added the demographic and anthropometric characteristics of the sample (sex distribution, mean age, height, weight, and BMI), as well as the statistical analysis information, to the Materials and Methods section of the Abstract.

> Abstract section (pg. 1, line 27):Materials and Methods: A randomized crossover design was used and thirty healthy young adults (age 21.9 ± 1.5 years; BMI 20.9 ± 1.8 kg/m²) performed AB and VM in randomized order.”

> Abstract section (pg. 1, line 34): “Repeated-measures analysis of variance and paired t-tests were conducted on the datasets.”

  1. In the results, in addition to presenting p-values, please include the means and standard deviation.

Response: In the Results section of the Abstract, we have added the mean ± SD(SE) values for the key outcomes of each intervention (PI and SpOâ‚‚), while the remaining outcome measures are described in detail in the Results section of the main text.

> Abstract section (pg. 1, line 35):Results: Both the VM and AB significantly increased heart rate (p < 0.001), and systolic blood pressure (VM: p = 0.015; AB: p < 0.001). Cerebral oxygen saturation decreased significantly (VM: p < 0.05; AB: p < 0.05), whereas oxyhemoglobin increased during both interventions, suggesting higher cerebral oxygen demand. The VM specifically increased the carotid pulsatility index (pre = 1.76 ± 0.28; post2 = 1.87 ± 0.33; p = 0.008), reflecting elevated central vascular resistance. In contrast, AB decreased peripheral oxygen saturation (pre = 98.43 ± 0.71; post1 = 97.49 ± 1.76; p < 0.001) and increased peripheral (heart–finger) pulse wave velocity (Lt: p = 0.026; Rt: p = 0.010), indicating greater stimulation of peripheral circulation.”

  1. In conclusion, focus on concluding only the proposed objectives.

Response: Following the reviewer’s advice, we have revised the Conclusion section in the Abstract to be more concise.

> Abstract section (pg. 1, line 43):Conclusions: Distinct mechanisms that elevate intra-cavity pressure differentially influence central and peripheral hemodynamics. These findings suggest that intra-cavity pressure can selectively modulate hemodynamic responses, with potential applications in both clinical and exercise settings.”

Keywords:

  1. Avoid repeating words already mentioned in the title, please replace them with synonyms.

Response: The term “abdominal bracing” was replaced with “isometric abdominal activation”. Moreover, “Valsalva maneuver” was substituted with “forced exhalation”, depending on the context.

> Keyword section (pg. 2, line 47): “isometric abdominal activation; forced exhalation; intra-abdominal pressure; intra-thoracic pressure; cerebrovascular circulation; peripheral circulation”

Introduction:

  1. Begin the introduction with epidemiological information on the topic, then begin presenting the study questions more specifically.

Response: Thank you for your valuable suggestion.

Peripheral arterial disease (PAD), affecting more than 200 million individuals globally, shows a substantial association with hypertension, with approximately 17.0% of cases attributable to elevated blood pressure in a large epidemiological study [2, added as reference 4].

Reflecting this evidence, additional epidemiological context has been incorporated into the Introduction section, where the physiological relevance and potential clinical applicability of each breathing intervention are presented, followed by a concise description of the study objective and primary research question.

[2] Criqui, M. H.; Aboyans, V. Epidemiology of Peripheral Artery Disease. Circulation Research 2015, 116 (9), 1509–1526. https://doi.org/10.1161/CIRCRESAHA.116.303849.

[4] Meijer, W. T.; Grobbee, D. E.; Hunink, M. G.; Hofman, A.; Hoes, A. W. Determinants of Peripheral Arterial Disease in the Elderly: The Rotterdam Study. Arch Intern Med 2000, 160 (19), 2934–2938. https://doi.org/10.1001/archinte.160.19.2934.

> Introduction section (pg. 2, line 58): “Peripheral arterial disease (PAD), affecting more than 200 million individuals globally, shows a substantial association with hypertension, with approximately 17.0 % of cases attributable to elevated blood pressure in a large epidemiological study [2,4].”

  1. Clearly present the study hypotheses, as well as the current state of the art and how this work will contribute to the evolution of the field.

Response: In accordance with the reviewer’s suggestion, the following content has been added to the Introduction section:

> Introduction section (pg. 2, line 91): "On the other hand, there remains a lack of research addressing hemodynamic changes induced by intra-cavity pressure alterations such as those produced during the VM and AB. Prior studies have also reported VM-induced cerebral hemodynamic alternations mediated by blood pressure and physiological changes [9,15,16]; however, these investigations have predominantly focused on the carotid artery, leaving peripheral vascular responses insufficiently explored. Similarly, limited evidence exists regarding the hemodynamic effects of AB. Although AB shares a mechanistic similarity with VM, previous studies have largely centered on muscle activation and core stability, with hemodynamic influences remaining under investigated.

Therefore, the present study aimed to examine the impact of distinct mechanisms for increasing intra-cavity pressure through the VM and AB on central and peripheral hemodynamics to evaluate their physiological significance and potential clinical applicability. Notably, this study represents the first attempt to compare VM and AB from a hemodynamic perspective, highlighting its potential academic relevance. Based on this rationale, the study hypotheses were as follows. (1) VM and AB would affect the structure and function of either central or peripheral vessels. (2) Differences in pressure induction between VM and AB would elicit distinct effects on the structure and function of central and peripheral vessels."

  1. At the end of the introduction, please adjust the objectives in the same way as those presented in the abstract.

Response: In accordance with the reviewer’s comment, the study objective will be added in the Introduction section in a form consistent with the Abstract, as follows:

> Introduction section (pg. 3, line 100): “Therefore, the present study aimed to examine the impact of distinct mechanisms for increasing intra-cavity pressure through the VM and AB on central and peripheral hemodynamics to evaluate their physiological significance and potential clinical applicability.”

Materials and Methods

  1. Begin writing the methodology by presenting the type of study and the location where the intervention was performed.

Response: Thank you for your valuable suggestion. The study type and intervention location have been included at the start of the Methodology section as follows.

> Materials and Methods section (pg. 5, line 208): “A prospective, controlled experimental study was carried out at the ultrasonography room, where all VM and AB interventions were performed in a controlled laboratory setting.”

  1. Present the technical information of the equipment used for the anthropometric assessment of the participants.

Response: Thank you for your valuable suggestions. The following content has been added to the revised manuscript: In this study, participants’ body mass index (BMI) was calculated using values obtained from baseline measurements of height and weight. Body weight and height were measured using a scale (Atflee T3; Atflee Co., Ltd., Seoul, Republic of Korea) and a flexible tape measure, respectively. Height was measured with participants standing barefoot, upright, and facing forward in the Frankfort plane [added as reference 18].

[18] Finch, H.; Arumugam, V. Assessing the Accuracy and Reliability of Direct Height Measurement for Use in Adult Neurological Patients with Contractures: A Comparison with Height from Ulna Length. Journal of Human Nutrition and Dietetics 2013, 27, 48–56. https://doi.org/10.1111/jhn.12103.

> Materials and Methods section (pg. 3, line 131): “BMI was calculated from baseline measurements of weight and height obtained using a calibrated scale (Atflee T3; Atflee Co., Ltd., Seoul, Republic of Korea) and a measuring tape, with height measured while participants stood barefoot, upright, and facing forward in the Frankfort plane [18].”

  1. Was the study registered as a clinical trial?

Response: This study was registered with the Clinical Research Information Service (KCT0009742; registered on 30 August 2024), and this information has been included in both the Abstract and the Materials and Methods sections.

  1. Please describe the statistical analyses in chronological order, including the normalization tests, how the data were presented, and which tests were used.

Response: Thank you for valuable comment. The statistical analysis techniques and data presentation methods used in this study have been described in detail within the Statistics section.

> Materials and Methods section (pg. 8, line 264): “Normality of all data was assessed using the Kolmogorov-Smirnov test, with p > 0.05 indicating a normal distribution.”

  1. Check the quality of the flowchart resolution; it is blurry.

Response: Thank you for your constructive feedback. The resolution of the original image was increased to 600 dpi, and the revised version has been re-uploaded.

Results:

  1. Begin reporting the results with the sociodemographic information of the study sample.

Response: Thank you for your valuable suggestion. The first paragraph of the Results section has been updated to include participants’ sociodemographic information, including age, ethnicity, highest educational level. Characteristics of participants already informed in the Materials and Methods section.

> Results section (pg. 8, line 267): “This study was conducted with thirty healthy participants. All participants were ethnically homogeneous and were university students. No participants withdrew from the experiment, and data from all thirty participants were included in the analysis.”

  1. Regarding t-tests, please also include information on the effect-size. Include this information in statistical analyses, tables, and/or in the body of the text.

Response: Following the reviewer’s recommendation, effect sizes, including Cohen’s d and partial eta squared, have been incorporated for all relevant variables in the revised manuscript.

Discussion:

  1. It's not clear to me why you're presenting figures in the discussion. Shouldn't this information be included in the results? Check and adjust.

Response: Thank you for your suggestion. The figure presented in the Discussion, originally used to illustrate key outcome measures, has been relocated to the Results section in accordance with the reviewer’s suggestion.

  1. At the end of the discussion, after addressing the limitations, please include the practical applications of the research.

Response: We appreciate your valuable suggestion. Based on the reviewer’s suggestions, brief mentions of the study results, their relevance to the study objectives, and the clinical implications have been added.

> Discussion section (pg. 12, line 434): “This study is novel in that few previous investigations have directly compared the differential hemodynamic effects of thoracic and abdominal pressure on central and peripheral parameters in healthy participants. It provides a scientifically in-depth perspective on the effects of these interventions on blood flow, complementing prior studies that primarily focused on muscle activation and functional strengthening. Future research should consider comparing various pressure intensities and evaluating the long-term effects of these interventions to further extend these findings.”

  1. Also, try to clarify in the discussion how this research will influence the state of the art on the topic.

Response: In accordance with the reviewer’s suggestion, the following content regarding the study’s novelty and future research directions has been added to the Discussion:

> Discussion section (pg. 12, line 434): “This study is novel in that few previous investigations have directly compared the differential hemodynamic effects of thoracic and abdominal pressure on central and peripheral parameters in healthy participants. It provides a scientifically in-depth perspective on the effects of these interventions on blood flow, complementing prior studies that primarily focused on muscle activation and functional strengthening. Future research should consider comparing various pressure intensities and evaluating the long-term effects of these interventions to further extend these findings.”

Conclusions:

  1. The conclusion is too long; please shorten it to address only the proposed objectives.

Response: Thank you for your valuable suggestion. In line with the comment from the previous reviewer, and following the reviewer's recommendation, we changed the Conclusion section as follows.

> Conclusion section (pg. 12, line 443): “In conclusion, although both interventions increased blood pressure, the VM predominantly influenced the central vascular system, whereas AB primarily affected the peripheral system. This distinction supports their selective application in exercise prescription, rehabilitation strategies, and blood flow assessment. Future studies should compare different pressure intensities and evaluate long-term effects to expand on these findings.”

  1. Regarding the bibliographic references, I found that many are more than 5 years old. Therefore, I suggest you replace older references with more recent ones, that is, from 2022 onward.

Response: Thank you for your opinion. In response to the reviewer’s suggestion, certain references have been replaced with more recent studies that provide supporting evidence for the same content as follows.

[1] Boppana, A.; Lee, S.; Malhotra, R.; Halushka, M.; Gustilo, K. S.; Quardokus, E. M.; Herr, B. W.; Börner, K.; Weber, G. M. Anatomical Structures, Cell Types, and Biomarkers of the Healthy Human Blood Vasculature. Sci Data 2023, 10 (1), 452. https://doi.org/10.1038/s41597-023-02018-0.

[12] Sembera, M.; Busch, A.; Kobesova, A.; Hanychova, B.; Sulc, J.; Kolar, P. The Effect of Abdominal Bracing on Respiration during a Lifting Task: A Cross-Sectional Study. BMC Sports Sci Med Rehabil 2023, 15, 112. https://doi.org/10.1186/s13102-023-00729-w.

[13]  Mitra, K.; Kunte, S. A.; Taube, S. E.; Sankarlinkam, S.; Mohamed, L.; Adodo, E.; Green, C. L.; Fudim, M.; Richardson, E. S. Standing under Pressure: Hemodynamic Effects of Abdominal Compression Type and Intensity in Healthy Adults. Front. Physiol. 2025, 16. https://doi.org/10.3389/fphys.2025.1621617..

[27] Babamohamadi, H.; Ameri, Z.; Asadi, I.; Asgari, M. R. Comparison of the Effect of EMLA™ Cream and the Valsalva Maneuver on Pain Severity during Vascular Needle Insertion in Hemodialysis Patients: A Controlled, Randomized, Clinical Trial. Evidence-Based Complementary and Alternative Medicine 2022, 2022 (1), 8383021. https://doi.org/10.1155/2022/8383021.

[45] Park, C.-S.; Kim, M.-J.; Kim, D.-H.; Lee, Y.-B.; Kang, C.-K. Functional Near-Infrared Spectroscopy Analysis of Cerebral Physiological Changes in Response to Atmospheric Gas Concentrations. Applied Sciences 2024, 14 (24), 11525. https://doi.org/10.3390/app142411525.

Round 2

Reviewer 1 Report

Comments and Suggestions for Authors

Dear autors,

I sincerely appreciate your kind consideration of our previous comments. However, I would like to raise a lingering concern: the phrase “participants were recruited who did not have cerebrovascular, neurological, or other medical conditions” actually describes exclusion criteria, not inclusion criteria.

Inclusion criteria specify who can participate in the study based on characteristics that define the target population: for example, age, sex, diagnosis, functional level, membership in a specific clinical group, or exposure to a particular intervention.

Author Response

Authors’ Responses to Reviewers’ Comments:

We sincerely appreciate your valuable comments and suggestions, which helped us improve the quality of our manuscript. We carefully revised the manuscript following the reviewer’s recommendations accordingly. Point-by-point responses are as follows:

Comments and Suggestions for Authors:

-----------------------------------------------------------------------------------------------------

Reviewer #1:

Dear authors,

I sincerely appreciate your kind consideration of our previous comments. However, I would like to raise a lingering concern:

Specific comments:

  1. The phrase “participants were recruited who did not have cerebrovascular, neurological, or other medical conditions” actually describes exclusion criteria, not inclusion criteria.

Inclusion criteria specify who can participate in the study based on characteristics that define the target population: for example, age, sex, diagnosis, functional level, membership in a specific clinical group, or exposure to a particular intervention.

Response: We appreciate your insightful comment. We clarified the inclusion criteria for participant recruitment in this section.

> 2. Materials and Methods section (pg. 3):Thirty healthy adults in their twenties were included, meeting the inclusion criteria of having a normal body mass index (BMI) and blood pressure, and being able to generate expiratory pressures of 20–25 mmHg.

Reviewer 2 Report

Comments and Suggestions for Authors

Dear Authors,

Thank you for considering my suggestions and incorporating them into the manuscript, which has been globally improved. Congratulations.

Below are some specific suggestions with line indications.

L29 – Please revise the uppercase and lowercase criteria [e.g. “abdominal bracing (AB) and Valsalva Maneuver (VM)]. Please consider this suggestion throughout the manuscript.

L248 – Subtitle suggested in full.

L293 – Please consider presenting Figure 2 after the introductory text.

L426 – Please consider presenting Table 1 after the introductory text.

L548-564 – Some paragraphs are too long, others too short. Please consider standardizing to improve readability (8-12 lines suggested).

End of discussion section. Before the suggestions for future research, please indicate the study limitations.

L668-672 - Conclusions – Please consider developing the conclusions section and placing the suggestions for future research at the end of the discussion section.

Please double-check the references format and English details.

Comments on the Quality of English Language

The English is globally with good quality. Some details can be improved.

Author Response

Authors’ Responses to Reviewers’ Comments:

We sincerely appreciate your valuable comments and suggestions, which helped us improve the quality of our manuscript. We carefully revised the manuscript following the reviewer’s recommendations accordingly. Point-by-point responses are as follows:

Comments and Suggestions for Authors:

-----------------------------------------------------------------------------------------------------

Reviewer #2:

Dear Authors,

Thank you for considering my suggestions and incorporating them into the manuscript, which has been globally improved. Congratulations.

Below are some specific suggestions with line indications.

Specific comments:
1. L29 – Please revise the uppercase and lowercase criteria [e.g. “abdominal bracing (AB) and Valsalva Maneuver (VM)]. Please consider this suggestion throughout the manuscript.

Response: We appreciate your insightful comment. “Valsalva” is an eponym and should be capitalized. In abstract, the term “Maneuver” corrected to lowercase as “maneuver”. For reference, [Appelboam, A. et al., 2015] also capitalizes “Valsalva” in the middle of the sentence. In contrast, “abdominal bracing” is not an eponym, so it is correctly written in lowercase when appearing in the middle of a sentence.

(1) Appelboam, A.; Reuben, A.; Mann, C.; Gagg, J.; Ewings, P.; Barton, A.; Lobban, T.; Dayer, M.; Vickery, J.; Benger, J. Postural Modification to the Standard Valsalva Manoeuvre for Emergency Treatment of Supraventricular Tachycardias (REVERT): A Randomised Controlled Trial. The Lancet 2015, 386 (10005), 1747–1753. https://doi.org/10.1016/S0140-6736(15)61485-4.

  1. L248 – Subtitle suggested in full.

Response: Thank you for your suggestion. We would like to clarify that we have revised the subtitle as “Peripheral Pulse Wave Velocity” according to your recommendation.

  1. L293 – Please consider presenting Figure 2 after the introductory text.

Response: Thank you for your valuable suggestion. To improve readability and enhance the reader’s understanding, we have changed Figure 2 to Figure 1 and repositioned it earlier in the manuscript. In addition, the figure is now placed immediately after its first mention in the text.

  1. L426 – Please consider presenting Table 1 after the introductory text.

Response: Thank you for your valuable suggestion. However, since Table 1 presents the main results, we believe it is more appropriate to include it in the latter part of the Results section.

Table 1. Simplified comparison of hemodynamic responses (pulsatility index, regional oxygen saturation and peripheral capillary oxygen saturation) during the Valsalva maneuver and abdominal bracing.

Repeated-measures comparison

Post hoc comparison (Tukey)

Group

Parameter

Variable

Mean ± SD

F

p

Repeated-measures factors

t

pTukey

VM

PI

Pre

1.76 ± 0.28

4.48

0.008*

Pre

Post2

–3.29

0.020*

Post1

1.82 ± 0.33

Post2

1.87 ± 0.33

Post3

1.84 ± 0.32

Post4

1.81 ± 0.30

rSO2 (Lt)

Pre

45.37 ± 13.09

5.70

0.005*

Pre

Post1

3.26

0.023*

Post1

41.12 ± 13.64

Post2

44.24 ± 14.03

Post3

43.61 ± 14.90

Post4

43.74 ± 14.27

rSO2 (Rt)

Pre

48.46 ± 13.78

10.99

<0.001*

Pre

Post1

4.06

0.003*

Post1

43.02 ± 14.91

Post2

47.68 ± 13.88

Post1

Post2

–3.23

0.025*

Post3

47.63 ± 13.82

Post3

–3.40

0.016*

Post4

47.86 ± 13.38

Post4

–3.53

0.012*

AB

SpO2

Pre

98.43 ± 0.71

5.56

<0.001*

Pre

Post1

3.17

0.027*

Post1

97.49 ± 1.76

Post2

97.68 ± 1.45

Post3

98.30 ± 0.75

Post4

98.30 ± 0.75

rSO2 (Lt)

Pre

44.32 ± 14.03

6.15

0.001*

Pre

Post1

3.47

0.014*

Post1

40.83 ± 15.09

Post2

41.66 ± 16.26

Post2

3.08

0.035*

Post3

42.30 ± 15.39

Post3

3.32

0.020*

Post4

42.60 ± 15.09

rSO2 (Rt)

Pre

48.74 ± 13.95

5.95

0.002*

Pre

Post1

3.36

0.018*

Post1

45.00 ± 15.48

Post2

46.58 ± 15.21

Post3

47.13 ± 14.62

Post4

46.70 ± 15.32

Abbreviations: VM, Valsalva maneuver; AB, abdominal bracing; PI, pulsatility index; rSO2, regional oxygen saturation; SpO2, peripheral capillary oxygen saturation; Lt, left; Rt, right; SD, standard deviation; * p < 0.05.

  1. L548-564 – Some paragraphs are too long, others too short. Please consider standardizing to improve readability (8-12 lines suggested).

Response: Thank you for your constructive feedback. We have adjusted the paragraph lengths to preserve their original flow while standardizing their structure as much as possible.

  1. End of discussion section. Before the suggestions for future research, please indicate the study limitations.

Response: We appreciate your insightful comment. In accordance with the reviewer’s suggestion, we have presented the discussion future research immediately following the study limitations.

> 4. Discussion section (pg. 12):Future research should consider comparing various pressure intensities, evaluating the long-term effects of these interventions, and including post-intervention follow-up with repeated interventions at different intensities to clarify sustained or cumulative effects.

  1. L668-672 - Conclusions – Please consider developing the conclusions section and placing the suggestions for future research at the end of the discussion section.

Response: Thank you for pointing this out. In accordance with the reviewer’s suggestion, we have further enhanced the Conclusions section as follows.

> 5. Conclusion section (pg. 12):In this study, the VM showed a significant increase in PI, which caused a transient rise in intrathoracic pressure, directly affecting the cardiovascular system and aorta. In contrast, AB led to decreases in SpO2 and peripheral PWV, reflecting and elevation in intra-abdominal pressure through abdominal core muscle contraction, resulting in greater peripheral muscle activation and oxygen consumption. In conclusion, although both interventions increased blood pressure, the VM predominantly influenced the central vascular system, whereas AB primarily affected the peripheral system. This distinction supports their selective application in exercise prescription, rehabilitation strategies, and blood flow assessment.

  1. Please double-check the references format and English details.

Response: Following the reviewer’s suggestions, we have carefully reviewed the reference format and English details to comply with the journal’s template requirements.

Reviewer 3 Report

Comments and Suggestions for Authors

Dear authors,

Thank you for providing the revised version of the manuscript. After carefully reviewing the adjustments made, I can see that the authors have met the reviewer's requirements and the manuscript has improved significantly.

Author Response

Authors’ Responses to Reviewers’ Comments:

We sincerely appreciate your valuable comments and suggestions, which helped us improve the quality of our manuscript. We carefully revised the manuscript following the reviewer’s recommendations accordingly. Point-by-point responses are as follows:

Comments and Suggestions for Authors:

-----------------------------------------------------------------------------------------------------

Reviewer #3:

Dear authors,

Thank you for providing the revised version of the manuscript. After carefully reviewing the adjustments made, I can see that the authors have met the reviewer's requirements and the manuscript has improved significantly.

Response: We thank the reviewer for the comments.
